# Simple ePDF: A Pair Distribution Function Method Based on Electron Diffraction Patterns to Reveal the Local Structure of Amorphous and Nanocrystalline Materials

**DOI:** 10.3390/nano13243136

**Published:** 2023-12-14

**Authors:** János L. Lábár, Klára Hajagos-Nagy, Partha P. Das, Alejandro Gomez-Perez, György Radnóczi

**Affiliations:** 1Thin Film Physics Laboratory, Institute of Technical Physics and Materials Science, HUN-REN Centre of Energy Research, Konkoly Thege M. út 29-33, H-1121 Budapest, Hungary; nagy.klara@ek.hun-ren.hu (K.H.-N.); radnoczi.gyorgy@ek.hun-ren.hu (G.R.); 2NanoMEGAS SPRL, Rue Èmile Claus 49 bte 9, 1050 Brussels, Belgium; partha@nanomegas.com (P.P.D.); alex@nanomegas.com (A.G.-P.)

**Keywords:** PDF, TEM, electron diffraction, disordered materials, glass, nanocrystals, nearest-neighbor distance, coordination number, decomposition, automation

## Abstract

Amorphous, glassy or disordered materials play important roles in developing structural materials from metals or ceramics, devices from semiconductors or medicines from organic compounds. Their local structure is frequently similar to crystalline ones. A computer program is presented here that runs under the Windows operating system on a PC to extract pair distribution function (PDF) from electron diffraction in a transmission electron microscope (TEM). A polynomial correction reduces small systematic deviations from the expected average Q-dependence of scattering. Neighbor distance and coordination number measurements are supplemented by either measurement or enforcement of number density. Quantification of similarity is supported by calculation of Pearson’s correlation coefficient and fingerprinting. A rough estimate of fractions in a mixture is computed by multiple least-square fitting using the PDFs from components of the mixture. PDF is also simulated from crystalline structural models (in addition to measured ones) to be used in libraries for fingerprinting or fraction estimation. Crystalline structure models for simulations are obtained from CIF files or str files of ProcessDiffraction. Data from inorganic samples exemplify usage. In contrast to previous free ePDF programs, our stand-alone program does not need a special software environment, which is a novelty. The program is available from the author upon request.

## 1. Introduction

The physical properties of materials critically depend on the arrangement and distances of the atomic species in the material in question. Structure models for crystalline materials are described in CIF files [1] or proprietary files, like the str files of ProcessDiffraction [2] (and self-references within it). Examinations of structure at the atomic level started with the discovery of X-rays and continued with the application of the wave nature of other particles, like neutrons and electrons. More than a century ago, Debye developed a formula to calculate the intensity scattered by an agglomerate of atoms [3]. Very soon, the formula was also applied to scattering experiments using electrons [4]. At that time (1930), the evaluation of experiments was limited to “manual” calculations. Calculation of atomic density in real space was obtained using a transformation of the measured intensities’ distribution [5]. The appearance of computers helped the calculation of more atoms in the examined volume. The procedure became especially efficient by introducing the Fast Fourier Transformation (FFT) in the processing of the Debye formula [6]. The first 100 years of that early development was professionally summarized in [7]. By the start of the 21st century, crystallography (the science of determining the structure of materials with long-range order) matured, and the structures of very complex materials (like crystallized proteins) were determined with the help of methods with a basis in (at least partially) the Debye formula.

However, a plethora of important materials do not possess long-range order. By increasing defect density, long-range order starts to diminish, and when breaking a crystal into small pieces, resulting in nanocrystalline powders, “long range” order only prevails in limited local surroundings. Using very fast vacuum deposition of atoms or quenching liquids, we arrive at amorphous or glassy materials, where only short-range order exists. Amorphous thin layers in semiconductor devices are key factors in device performance. Bulk amorphous solids (typically metal-based) have very different (enhanced) mechanical properties compared to their crystalline counterparts. The solubility (and consequently efficiency) of amorphous active pharmaceutical components is an order of magnitude higher than that of the crystalline form of the same chemical compound. Consequently, examination of the local structure of disordered materials is a very important field.

Among others, X-ray diffraction (XRD)-based radial distribution function (RDF) analysis showed the densification of amorphous Si (a-Si) [8]. Cockayne and co-workers showed that electron diffraction (ED)-based RDF analysis can also be applied to refine structural models [9]. Farrow and Billinge described the relationship between PDF and small-angle scattering [10]. Weber and Simonov extended PDF analysis to disordered single crystals by introducing three-dimensional PDF [11]. Schmidt and coworkers combined 3D electron diffraction from single-crystal diffuse scattering with 3D difference PDF and obtained good results that they compared to XRD and neutron diffraction results [12]. Modelling of RDF from structural data was also developed for organic materials [13]. Mitchell and Petersen published a useful free software (called RDFTools) that performs RDF-related calculations from electron diffraction data and runs under a Digital Micrograph™ (DM) (Pleasanton, California, Gatan) environment [14]. The alternative ePDF Tools program also needs a DM environment [15]. Another free program for electron diffraction PDF calculation (SUePDF) runs in the MATLAB environment [16]. eRDF Analyser is another software tool available in MATLAB for evaluating electron diffraction data [17]. Mu and coworkers mapped amorphous phases from 4D-STEM datasets [18] where a 2D electron diffraction pattern was recorded at each pixel of a 2D scanning transmission electron microscope (STEM) image (4D-STEM). They demonstrated the operation of their method on organic molecules. Later, they combined PDF analysis with independent component analysis to automatically classify the amorphous components [19]. Unfortunately, it is sometimes difficult to find a relationship between a principal component and a true thermodynamic phase. Liu and coworkers applied non-negative matrix factorization (NMF) for rapid automatic classification of PDFs obtained in in situ synchrotron experiments. They compared NMF and principal component analysis and showed that NMF yields physically meaningful PDF components [20]. Mu and coworkers also used energy-filtered electron diffraction and pinpointed that the Q-dependence of the measured electron scattering deviates from that of the calculated average scattering and corrected for the difference by removing a polynomial function from F(Q) [21]. This shape deviation was attributed to dynamic scattering by several authors as the result of multiple scattering of electrons. Anstris and coworkers suggested that multiple scattering should be corrected for by a log-normal procedure, similarly to that applied in EELS and EDS [22]. Although it appears to be an attractive solution, reports of variable success appeared in the literature. Rakita and coworkers mapped a bulk metallic glass (BMG) with 4D-STEM and with the help of machine learning they separated partial PDFs for the modelled quasi-binary alloy [23]. They named their approach scanning nano-structure electron microscopy (SNEM) [23]. Junior and coworkers determined the structure of ice with ePDF in cryogenic TEM and compared it to the structure determined by PDF based on XRD and neutron diffraction [24]. The program PDFgetX3 is for PDF analysis of XRD data [25]. A program called ePDF suite is available commercially [26]. Gorelik and coworkers published a concise summary of the development and state of ePDF analysis [27]. They also presented a list of available PDF programs. Since the freely distributed programs require a special software environment (DM or MATLAB), a need remains for a free stand-alone program that does not require any other software environment. The present publication is about such a stand-alone ePDF implementation. Functionalities are described in Chapter 2, while additional information about the program is presented in Appendix A.

The first version of the ProcessDiffraction program was mainly used for qualitative phase analysis of nanocrystalline samples by comparing the 1D distribution to Marker lines calculated from model structures. Calculation of different PDF functions was also incorporated into ProcessDiffraction and applications of these PDF functionalities appeared in several publications [28,29,30,31]; however, this part of the program has never been published. Simple ePDF also contains a special non-linear least-square routine developed for diffuse broad rings of amorphous samples to find their center and correct for their elliptical distortion [32]. Since ePDF analysis is separate and different from the published parts (indexing of single crystals, quantitative phase analysis of nanocrystalline powders, calculating phase and orientation maps from 4D-ED data and calculating strain distribution from 4D-ED data), we developed Simple ePDF as a stand-alone program separated from ProcessDiffraction and elaborate its operation here. Functionalities, which are also used in Simple ePDF from the earlier developments (handling of crystal structures, generation of Markers, usage of Mask, extending dynamic range) are not elaborated here but can be found in the references of [2].

Although ePDF analysis is mainly valid for amorphous samples, it also gives a very good approximation for an assembly of nanocrystals. Grain size (for nanocrystalline samples) and cluster size (for amorphous samples) will be used interchangeably in this paper.

## 2. Functionalities of the Program

### 2.1. From Measured 2D Pattern to Distribution of Atom Pairs in Real Space

A typical 2D electron diffraction pattern from a nanocrystalline (Figure 1a) or amorphous (Figure 1c) sample contains concentric diffuse broad rings of intensity, with the rings becoming sharper and sharper with increasing crystallinity. In many examples, the intensity enhancement along rings is only minor as compared to the smoothly decreasing large background [32]. Additionally, the rings may have slight elliptical distortion due to the presence of electromagnetic lenses in the TEM. This is why finding the center of these rings and correcting for the distortion needs a special non-linear fitting procedure, which is elaborated in [32]. The same routine is included in Simple ePDF. The intensity, *I*(*Q*) is plotted in a 1D curve, obtained by averaging intensity over the elliptically corrected rings in Figure 1b,d. Figure 1b,d also illustrate the effect of Masking in linear scale, where mainly large changes at low *Q* are observable (the smaller changes at high Q are shown in log-scale Section 2.2). (*Q = 4π·sin*(*Θ*)*/λ*, where *λ* is the wavelength of the illuminating electron beam.) Such 1D intensity distributions, *I*(*Q*)s are the starting data for ePDF analysis.

Calibration of the measurement (1/nm/pixel) is also important, since at later stages of processing the measured distances between atomic pairs will linearly depend on that value. Calibration is performed prior to the measurement of the amorphous sample by recording a sharp ring pattern from a polycrystalline (preferably nanocrystalline) powder sample and calibrating the peaks in the 1D intensity distribution by Markers deduced from known crystalline samples as in [2] (and self-references within it). The amorphous sample must be measured under identical experimental conditions. Reproducibility (e.g., precision) of this calibration is 0.3% relative [2].

Whenever there is a supporting thin film under the unknown amorphous sample, its effect can be approximately removed by subtracting the 1D intensity distribution of the supporting film (measured under identical dose and all other parameters) from that of the (support + unknown) combined sample. Among many other operations, such subtraction is easily achieved by the “Distribution Math” functionality of the program (see Section 2.10
*Utilities*).

When the allowed dose is limited due to beam-sensitive samples, the statistics can be improved by recoding diffraction patterns from physically identical parts of the sample and cumulating (the summation of) these patterns in their 2D forms prior to converting to 1D intensity distribution. This procedure is more favorable than summing the 1D intensity distributions.

PDF analysis needs diffraction patterns measured to as high a *Q*-range as possible. In addition to reducing the camera length, we can also extend the *Q*-range by positioning the electron beam close to one corner of the camera (in contrast to the usual position close to the center of the camera). It was proved in [32] that correct and accurate processing can also be reached with this beam position.

The naming conventions and Equations (1)–(5) follow definitions in [33]. The measured intensity distribution, *I*(*Q*), is first transformed into the reduced interference function *F*(*Q*) by removing the Laue monotonic scattering and scaling with the average scattering, both of them calculated from the atomic elastic scattering factors *f_i_*(*Q*), where the index *i* denotes the atomic species.
(1)$$F\left(Q\right)=Q\cdot \frac{I\left(Q\right)-N\cdot {\langle f_{i}\left(Q\right)\rangle }^{2}}{N\cdot \langle {f_{i}\left(Q\right)}^{2}\rangle }$$

*N* is a normalization constant that depends on the number of atoms in the scattering volume and the dose of electrons in the examined volume of the TEM sample. The angle bracket <> denotes averaging for the atomic species present. Simple ePDF offers two options for calculating *f_i_*(*Q*). The parameterization of Weickenmeier and Kohl [34] calculates the atomic elastic scattering factors *f_i_*(*Q*) for atomic numbers Z > 2 while the parametrization of Jiang and Li [35] is also valid for hydrogen. The *Pair Distribution Function* (PDF), *G*(*r*), is obtained from *F*(*Q*) using Fast Fourier Transformation (FFT):(2)$$G\left(r\right)=\frac{2}{\pi }\cdot \int_{Q_{min}}^{Q_{max}}F\left(Q\right)\cdot sin\left(Q\cdot r\right)\cdot dQ$$

*G*(*r*) gives the weight that the separation distance between a pair of atoms is *r*. The functional form (2) of the transformation assumes that the examined volume is isotropic. *G*(*r*) can also be expressed with the local deviation of *number density*, *ρ*(*r*) from the average value *ρ*_0_.
(3)$$G\left(r\right)=4\pi \cdot r\cdot \left[\rho \left(r\right)-\rho_{0}\right]$$

The information in PDF can also be presented in two related functions, which can be converted into each other back and forth. *g*(*r*) is the *Normalized Pair Correlation Function* (4), which gives the relative oscillations around the average density and the *Radial Distribution Function* (5), *RDF*(*r*), which gives the number of atoms in a spherical shell with radius *r*.
(4)$$g\left(r\right)=\frac{\rho \left(r\right)}{\rho_{0}}\;.$$
(5)$$RDF\left(r\right)=4\pi \cdot r^{2}\cdot \rho \left(r\right)=\;4\pi \cdot r^{2}\cdot \rho_{0}\cdot g\left(r\right)=r\cdot G\left(r\right)+4\pi \cdot r^{2}\cdot \rho_{0}.$$

Figure 2 gives the different forms of PDF calculated from data in Figure 1b.

The individual peaks in *G*(*r*) indicate an increased probability of atom pairs separated by distance *r* from each other. Slight shift and/or splitting of a peak indicate changes in local structure. By examining the local structure of geological samples of pumice and obsidian, we revealed the presence of two glass structures in amorphous silicic volcanic glasses [30]. The difference between the two was the Si-Si distances, showing the different connectivity of tetrahedra, which are also the basic units of structure in these amorphous samples. 

In another study, we examined soot particles from different atmospheric samples. Systematic comparison of the variation of peak positions and peak widths in reference to carbon compounds with different hydrogen content allowed us to propose that hydrocarbons are present in atmospheric soot particles in the form of small-sized aromatic moieties [29].

### 2.2. Expected Shape of F(Q) and an Empirical Correction

The *F*(*Q*) function is expected to oscillate around zero with decreasing amplitude at increasing *Q*-values. Both multiple scattering (from thicker samples) and unwanted stray radiation (together with signal intensities exceeding the linearity range of a camera) may cause deviation from this shape. In addition to the true distortions caused by scattering, the recorded pattern may also be distorted by the presence of shadowing elements (e.g., Beam-Stop) in the TEM in front of the camera. The effect of the latter is eliminated by Masking [32]. The effect of Masking on *I*(*Q*) is illustrated in Figure 3a. The logarithmic scale plot demonstrates that both the low-*Q* part and the high-*Q* part are influenced by shadowing elements and their effect is removed by the application of the Mask. 

When the high-*Q* tail of *F*(*Q*) significantly deviates from zero (in contrast to approaching it), we need an empirical correction. The empirical correction is embodied in the removal of a 4th-order polynomial to compensate for the systematic differences between the calculated *f_i_*(*Q*)s and the true shape of the average trend in *I*(*Q*). Figure 3b compares variations in the shape of *F*(*Q*) when it is computed from non-masked and masked *I*(*Q*)s either by relying on empirical correction or disregarding it. Figure 3c,d compare *G*(*r*)s calculated from the 4 variants of *F*(*Q*) in Figure 3b in a restricted *r*-range to enhance the visibility of differences. One can see that whenever the shape of *F*(*Q*) deviates from the expected form (i.e., the high-*Q* part increases (in Figure 3b) in contrast to the expected oscillation around zero), false oscillations appear in *G*(*r*) (Figure 3c,d). The conclusion that the additional oscillations are false are supported in Section 2.8
*Decomposition*, where the measured *G*(*r*) is compared to a combination of computed ones. (Section 2.8 shows that simulated composite *G*(*r*) only contains the same oscillations and no more as the measured masked, empirical distribution shown in Figure 3d.) 

We can conclude which one of the four experimental *G*(*r*)s in Figure 3c,d contain fewer artefacts by comparing these experimental ones to the simulated one in Section 2.8. We can see that the simulated version contains the same oscillations (and nothing else) as the one obtained from the masked version of the 2D diffraction pattern when we transform it into *F*(*Q*) using the empirical correction. This is a strong indication that the additional oscillations in the other three variants of *G*(*r*) are artefacts resulting from both the distortions caused by shadowing elements in the 2D pattern and the limited *Q*-range (due to too large of a camera length) together with the systematic differences between the shapes of measured and calculated atomic scattering (too thick of a sample). The empirical correction (of the form of a 4th-order polynomial) approximately corrects for the systematic differences between the shapes of the measured *I*(*Q*) the calculated atomic scattering factors ${\langle f_{i}\left(Q\right)\rangle }^{2}$ in formula (1). Stray scattering is one of the sources that can result in anomalous increases in intensity with increasing *Q*-values. Another source is double (multiple) scattering (as discussed, e.g., in [27]). Both of those sources result in a smoothly changing false intensity component, without the oscillations we look for in *F*(*Q*). That is why polynomial correction can remove the majority of those empirically, without the knowledge of their exact *Q*-dependence, leaving the oscillations in *F*(*Q*) less disturbed.

Section 2.9, *Back transformation to reciprocal space*, illustrates what modifications in the measured quantities (*F*(*Q*)’ instead of *F*(*Q*) and *I*(*Q*)’ instead of *I*(*Q*)) would correspond to the different corrections (masking on the one hand and the 4th-order polynomial empirical correction on the other hand) that we apply during processing.

### 2.3. Applying Physical Constraint to G(r) and Its Effect; Density and Coordination Number

Due to atomicity, we know that a pair of atoms cannot be closer to each other than a minimal distance *r*_0_, i.e., *ρ*(*r*) = 0 if *r < r*_0_. Using this fact, the form of Equation (3) dictates that *G*(*r*) must be a linear function with a slope of −4*πrρ*_0_ if *r < r*_0_. Deviation from this form is caused by unwanted effects in the measured *I*(*Q*) in addition to the limited *Q*-range of the measurement. We can correct for these problems by forcing a physically meaningful constraint (a linear function with a slope of −4*πrρ*_0_) on the starting part of *G*(*r*). With this physical constraint, we can get rid of many false oscillations at the start of *G*(*r*). This is illustrated in Figure 2d (to see what changes are needed in the original measured *F*(*Q*) to give the expected linear start in *G*(*r*), consult Section 2.9, *Back transformation to reciprocal space*). The slope, or in other words the value of number density, *ρ*_0_ is important since the measured value of the number of atoms in a given spherical shell (i.e., the coordination number) directly depends on the value of *ρ*_0_, as one can see looking at Equation (5).

Coordination numbers are obtained by integrating the RDF between two endpoints of a peak (endpoints of a peak (for deducing the coordination number) are best selected on *g*(*r*), since it oscillates around 1 for large r, in contrast to our integrand, *RDF*(*r*), which also contains a monotonically increasing parabolic component). Since the coordination number strongly depends on the number density, the latter must be determined as accurately as possible. The best situation is if the measured data are good enough that the start of *G*(*r*) is close to the expected linear function. In that case, the enforced linear curve is selected to join almost continually to the measured *G*(*r*) at *r*_0_ and the measured *G*(*r*) oscillates around the selected linear function if *r < r*_0_ (as in Figure 2d). The measured value of *ρ*_0_ is deduced from the slope. 

In a good example, when it was possible to determine the number density from the measured data reliably, the predecessor of our Simple ePDF program (unpublished version of ProcessDiffraction) was used in a study to decide which local order prevails in amorphous FeS produced under different systematic transformation conditions [31]. In the four related crystalline structures, either tetrahedral or octahedral local order existed. With the help of ePDF analysis, it was concluded that under the examined conditions, tetrahedral local order occurred in the amorphous FeS [31].

When the distorted shape of the measured data makes the determined *ρ*_0_ unreliable, we need to rely on an alternative approach, and we need to assume the value of *ρ*_0_ from another source. In such cases, the linear function is drawn with the known slope, which gives a density value different from the expected. It is interesting to note that the neighbor distances are not affected noticeably with the change in number density. The way to enforce a new density value without modification of the shape of *G*(*r*) is scaling *G*(*r*) with a constant. As an example, the coordination number of the first atomic shell is calculated for Mn82Cu3O15. The number density is *ρ*_0_ = 52 atoms/nm^3^ if the raw transformed *G*(*r*) is used. The coordination number calculated with this density is 6.8, far from the close packed value of 12. The true number density in crystalline alpha-Mn is *ρ*_0_ = 82 atoms/nm^3^. The coordination number calculated with this density is 11.8, very close to the value expected for nanocrystalline alpha-Mn.

### 2.4. Pearson’s Correlation Coefficient and Fingerprinting

We frequently face a problem of determining how much a measured local structure resembles a reference. Such similarity is expressed by Pearson’s correlation index, *p_mr_*, which is calculated for a limited interval of *G*(*r*) measured at discrete locations *r_i_*
(6)$$p_{mr}=\frac{\sum_{i}\left[\left\{G_{m}\left(r_{i}\right)-\bar{G_{m}}\right\}\cdot \left\{G_{r}\left(r_{i}\right)-\bar{G_{r}}\right\}\right]}{\sqrt{\sum_{i}\left\{G_{m}\left(r_{i}\right)-\bar{G_{m}}\right\}\cdot \left\{G_{m}\left(r_{i}\right)-\bar{G_{m}}\right\}}\cdot \sqrt{\sum_{i}\left\{G_{r}\left(r_{i}\right)-\bar{G_{r}}\right\}\cdot \left\{G_{r}\left(r_{i}\right)-\bar{G_{r}}\right\}}}$$
where the index *m* denotes “measured” and index *r* corresponds to “reference”, while the vinculum denotes the average over the given interval. The value of *p_mr_* is between 1 and −1, where 1 shows perfect correlation and −1 perfect anticorrelation. The 0 between corresponds to no correlation at all.

The reference PDF curves can be measured ones or simulated ones (showing the local structure of related crystalline materials). When a given measured PDF is compared to every member of a *G*(*r*) library, we call this action fingerprinting. The result is a list of *p_mr_* coefficients relating to the *rth* reference (1 ≤ *r* ≤ *N_ref_* and *N_ref_* is the number of references). The list is ordered in decreasing value of the Pearson coefficients, starting with the highest similarity. Table 1 is an example of the result of such Fingerprinting.

### 2.5. Simulating G(r) from Model Structures; Peak Width Determination and Grain Size Approximation

When we simulate the pair distribution function from a model (a list of atomic coordinates), we need to distinguish the different types of atoms and sum over all types of atoms in a spherical shell. The type of radiation (which is present in the measurement) comes into the calculation through the atomic scattering amplitudes, *f*.
(7)$$RDF\left(r\right)=\frac{1}{N{\langle f\left(0\right)\rangle }^{2}}\sum_{i}\sum_{j}{f\left(0\right)}_{i}{f\left(0\right)}_{j}\delta \left(r-r_{ij}\right)$$
and
(8)$$g\left(r\right)=\frac{RDF\left(r\right)}{4\pi r^{2}\rho_{0}}$$
where $r_{ij}$ is the distance between the central *i*-type atom and the *j*-type atom. *G*(*r*) is calculated from *g*(*r*) with Equation (3). Since usually we have the atomic coordinates for the unit cell in crystalline materials, we can easily simulate the *G*(*r*) for crystalline materials. Using the exact distance values from a crystalline model, our histogram of distances will be formed from delta functions. 

To obtain a simulated *G*(*r*) that can be compared to a measured one with finite peak width, we need to convolute the delta functions with a Gaussian function. That width must be identical to the width of a separated peak in the measured *G*(*r*) and we can determine it by non-linear-fitting a Gaussian function to that experimental peak. A peak with the shape of a Gaussian function is fitted to the top third of a separated measured peak by the Levenberg–Marquardt method [36]. That width is in reciprocal relation to the extent of measured oscillations in *F*(*Q*), which comes from the general properties of Fourier transformation.

With crystalline *G*(*r*)s in a library, we can search for fingerprints of local atomic arrangements in the amorphous material that resembles that of related crystalline materials. An approximation of grain size can be obtained by performing a quick calculation for the crystalline variant (producing oscillations all along *r → ∝*) and approximate grain size effects by introducing damping of the amplitudes in *G*(*r*) with increasing *r* in a second step. Such damping also has the effect of noise reduction in real space. An example of approximation of grain size by damping operation is shown in Figure 4.

### 2.6. Detecting Differences

Amorphous SiO_x_ films prior to and after irradiation with 20 MeV electrons were compared with our ePDF analysis [37]. The optical homogeneity of the layers was proved by ellipsometry. The homogeneous amorphous nature of both films was shown with high-resolution transmission electron microscopy (HRTEM). No separate clusters were identified by HRTEM. In contrast to the diffraction of nanocrystalline materials, where the formation of nanoclusters with different crystallographic structure changes the sharper ring structure, the formation of different amorphous nanoclusters in an amorphous matrix does not result in easily identifiable changes in the diffuse ring structure. However, the normalized pair distribution function, *g*(*r*), provides evidence for the changes in the atomic structure. We subtracted the *g*(*r*)s of the two samples from each other and showed that irradiation caused a decrease in the weight of Si-O distances and an increase in Si-Si distances (three (and only three) positive peaks appeared in the difference distribution (Figure 3b in [37]), which exactly corresponded to the first three atomic neighbor distances in crystalline Si. The nearest atomic distances in a-Si does not change from their crystalline counterpart due to the strong covalent bonding in Si). The changes in the ratio of Si-O and Si-Si bonds prove that amount of the elemental Si clusters increased in the sample. By incorporating the presence of Si clusters into the optical model of ellipsometry, the changes in optical absorption and refractive index were successfully explained [37].

Such difference analysis is only feasible in samples where the peaks in the measured *G*(*r*) are well separated. If several peaks of different *G_ij_*(*r*) partial components overlap, they can hide the changes or make identifying impossible.

### 2.7. Effect of Qmax

Since *G*(*r*) is calculated by FFT from *F*(*Q*), the discrete channel contents in the measured *I*(*Q*) (or better to say in *F*(*Q*), which is linearly related to them) are Fourier components of *G*(*r*). That is why oscillations with higher and higher *Q* result in better resolution (narrower peaks) in *G*(*r*). The extent of oscillations in the measured diffraction depends on cluster size (grain size). However, they may be truncated if the selected camera length in the TEM is too high. Consequently, the camera length should be low enough to record all the oscillations, which are present due to physical reasons. This effect can be best illustrated with nanocrystalline materials, where oscillations extend to higher *Q*-values (than in amorphous materials) due to the medium-range order. The recording medium (which practically means a camera in the present day) can also pose a limit to useful *Q_max_* in the recorded diffraction, because information may be lost in the noise for the lowest intensities (highest *Q*-values) if the maximum intensity (at low *Q*) is limited by the dynamic range of the camera. The limiting effect of noise illustrated in Figure 5a–c proves that the information content in the common intervals of *F*(*Q*)s recorded with different camera lengths is the same; however, we can extend the useful information by using lower camera lengths. Figure 5d shows how the resolution of the pair correlation function *G*(*r*) is improved with increased *Q_max_*. One can also see in Figure 5d that a too-short *Q*-range can lead to small shift of peaks and even changes in the number of detected peaks (false oscillations) in *G*(*r*). These false oscillations are also called termination ripples. It also demonstrates that when *Q_max_* is high enough, a further increase only makes a very little change in *G*(*r*).

### 2.8. Decomposition: Simple Mixtures: Estimating Fractions in Real Space; Surface Oxidation

If two (or more) amorphous (or nanocrystalline) phases are mixed in tiny clusters in such a way that none of the neighbor distances change in any of the clusters (no compound formation), we obtain a true mixture. (Mixture formation is not always intentional. When the surface of a TEM lamella is oxidized during transfer from preparation to the microscope, the excited volume is a mixture of the originally prepared material (in the body of the lamella) and the surface oxide. Any other surface contamination acts similarly.) We can synthetize the composite *G*(*r*) from the *G_j_*(*r*)s of the components for such true mixtures. Conversely, we can decompose the measured *G*(*r*) into the *G_j_*(*r*)s of the components with linear least-square fitting if the component *G_j_*(*r*)s are available either from measurements or from model calculations. The result is the fractions of the components, and we can compare the measured and the synthetized *G*(*r*)s as shown in Figure 6. It is important that we select in advance which known crystalline structures to include as components in the procedure to fit the local structure (in *G*(*r*)) of our nanocrystalline or amorphous sample. To illustrate the procedure, reference *G*(*r*)s were calculated from crystalline structures for alpha-Mn, fcc-Cu and MnO, and the *G*(*r*)s measured from Mn82Cu3O15 and Mn29Cu63O8 were decomposed into these 3 components. 

EDS analyses performed simultaneously with recording of the diffraction patterns provided the compositions of the samples. The knowledge of composition must be supplemented with information from other sources to facilitate reasonable selection of reference structures to be used in decomposition. Selection of phases generally relies on materials science considerations of the system, information from other measurements (e.g., X-ray photon spectroscopy, XPS, which indicates chemical bounds) or trial-and-error type selection of possible phases (with relevant compositions) from a structure database. In our case, alpha-Mn was selected (from several pure Mn structures) to fit a measured (and partially oxidized) Mn100Cu0 sample (not shown here). Oxygen content monotonically changed with Mn, so oxidation of Mn was assumed (in contrast to oxidation of Cu). The structure of MnO (out of several manganese–oxide structures) is a result of trial-and-error. Our decomposition procedure for the two samples in Figure 6 resulted in 82%Mn + 8%MnO + 10%Cu and 24%Mn + 4%MnO + 72%Cu, respectively. The *G*(*r*)s synthetized with these weights are shown in Figure 6. Although the agreement is not perfect, we can use it to follow trends in an approximate way. A detailed examination of systematic structure changes as a function of composition (in the entire range between pure Mn and pure Cu) from the point of structure evolution is the topic of a separate paper. 

### 2.9. Back Transformation to Reciprocal Space

The effect of constrained atomicity on the original reduced interference function *F*(*Q*) can be checked using the inverse FFT (iFFT) of the linearized *G*(*r*) (with a forced linear function at *r ≤ r*_0_). Figure 7a gives an example of how the originally measured *F*(*Q*) is related to *F*’(*Q*), obtained with the iFFT of the linearized *G*(*r*). We can see that false oscillations at high *Q* are reduced and the main change is in the low-*Q* region, close to the direct beam. This is an indirect indication that the false oscillations are strongly related to scattering from the Beam-Stop and from double scattering in the sample.

We can go one more step further and calculate back what shape the measured *I*(*Q*) should have been to result in *F*’(*Q*) during the processing. Reverse application of Equation (1) provides this *I*’(*Q*). Figure 7b compares the true measured *I*(*Q*) to the *I*’(*Q*) that resulted from the back-calculated *F*’(*Q*). The main difference between the *I*(*Q*) and *I*’(*Q*) is the average slope. It reflects the fact that the calculated average scattering decays faster with *Q* than the measured one. That difference is corrected empirically with the removal of a fitted polynomial. The enhanced intensity at higher *Q*-values is attributed to double (multiple) scattering within the sample. 

### 2.10. Utilities

To facilitate the generation of synthetic *G*(*r*)s, the program offers some utilities for ***real space*** functions. The operator can copy *G*(*r*) distributions to different memories, scale *G*(*r*)s together to see variations better, multiply the distribution by a constant and can subtract one *G*(*r*) from another. By sequential application of these functionalities, one can approximate a simulated *G*(*r*) for a mixture from the components.

An even wider range of utilities is available in ***reciprocal space*** for manipulating 1D distributions of diffracted intensities. They can be added, subtracted and multiplied by a constant, and two distributions can be multiplied together and can also be Fourier transformed.

## 3. Conclusions

In contrast to previous free ePDF programs, SimpleePDF runs under the Windows operating system as a stand-alone program which does not require any special programming environment. It is divided into three logical blocks, each maintaining simple operation with a minimal number of menu points and possible choices. At each step, the entire project can be saved and continued from that point after interruption. Conversion of the measured 2D electron diffraction pattern into 1D distribution, its correction and calibration are performed as in the previously published ProcessDiffraction program. Simple ePDF starts from those saved 1D distributions. *F*(*Q*) is computed with optional empirical correction. *G*(*r*) is calculated by FFT and the atomicity condition is enforced on it. Atomic distances and coordination numbers are determined and differences in *g*(*r*) can prove and visualize the changes and development of the local structure (between differently treated samples). The local structure can be compared to that of known crystalline phases whose structure is stored in CIF or str structure files. *G*(*r*) can be simulated from the same structure files. Fingerprinting automatically compares *G*(*r*) to a list of Reference *G*(*r*) files. For true mixtures (with no interaction between the clusters) even a semi-quantitative decomposition into suspected components is offered. The ePDF method (implemented in this program) is a useful tool for structure examination of either amorphous or nanocrystalline systems or a mixture of them and can be used in diverse fields of materials science.

## Figures and Tables

**Figure 1 nanomaterials-13-03136-f001:**
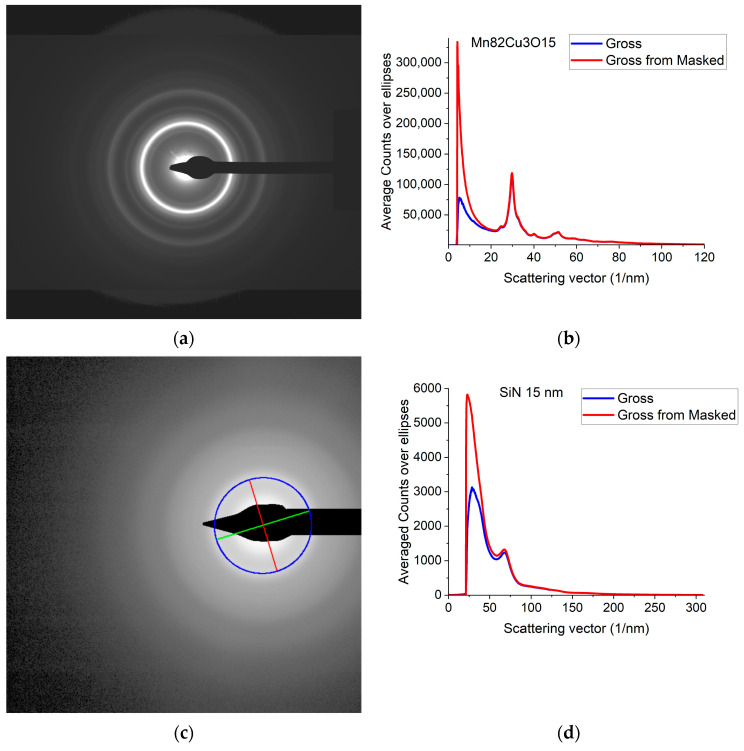
Electron diffraction pattern from nanocrystalline Mn82Cu3O15 (composition determined by EDS) and from amorphous SiN thin films. (**a**) 2D pattern as recorded by the camera from nanocrystalline Mn82Cu3O15; (**b**) 1D intensity distribution, *I*(*Q*) obtained by averaging the intensity over the elliptically corrected rings. The Beam-Stop is shown to demonstrate the effect of Masking [32]; (**c**) 2D pattern from amorphous SiN. The blue circle and the two lines aid centering and are visualized optionally; (**d**) 1D *I*(*Q*) obtained from (**c**).

**Figure 2 nanomaterials-13-03136-f002:**
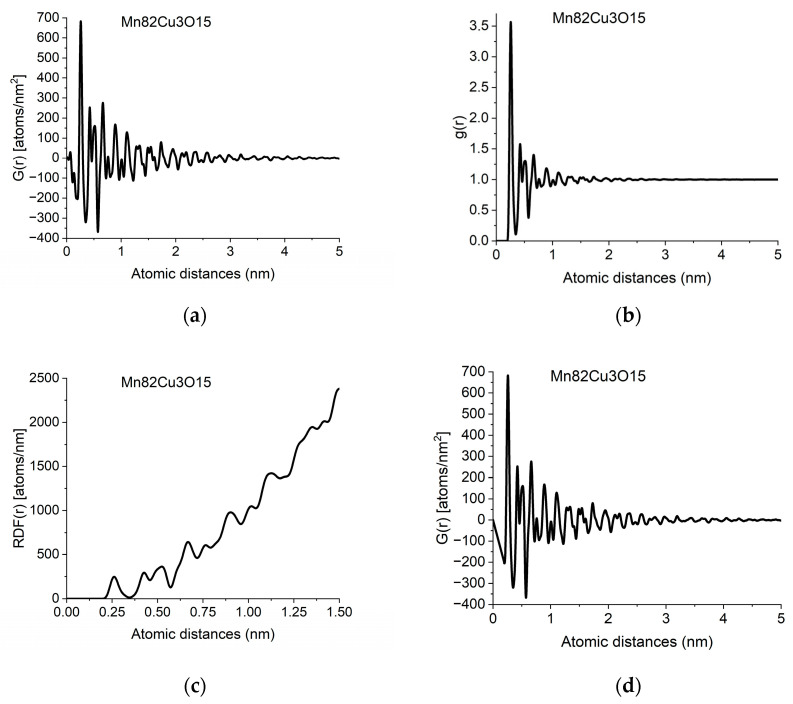
Different forms of PDF calculated from data in Figure 1b: (**a**) pair distribution function, *G*(*r*); (**b**) normalized pair correlation function, *g*(*r*); (**c**) radial distribution function, *RDF*(*r*); (**d**) effect of applying the physical constraint of atomicity to (**a**) (see Section 2.3).

**Figure 3 nanomaterials-13-03136-f003:**
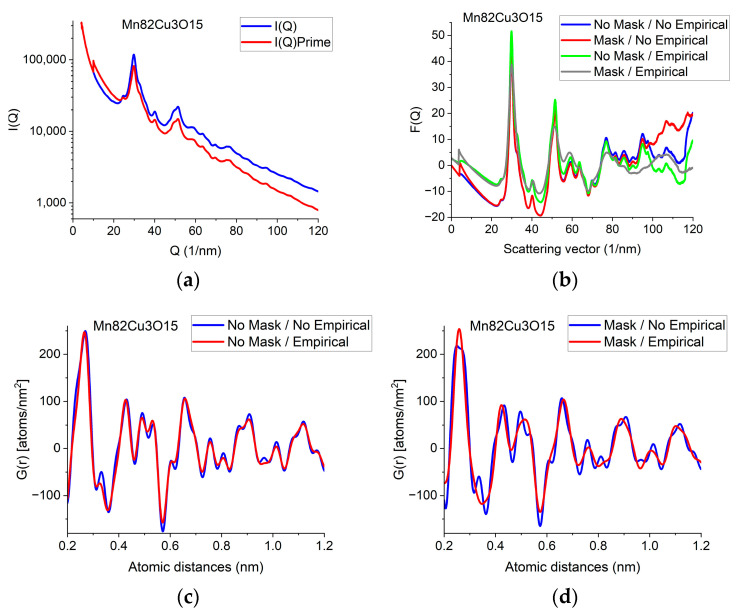
Illustration of how artefacts in the measured data deviate the shapes of *F*(*Q*) and *G*(*r*) from the expected ones and the effect of an empirical correction (removal of a 4th-order polynomial). (**a**) Shapes of *I*(*Q*) measured for Mn82Cu3O15 are compared when Mask is applied or not. The logarithmic scale simultaneously enhances the differences at low and high *Q*. At high *Q*, the shadow of the numbering unit is masked out. (**b**) Shapes of *F*(*Q*) are compared when calculated from the experimental *I*(*Q*)s in (**a**) either with or without Mask and either with or without empirical correction. (**c**) Effect of empirical correction when the source is the measured *non-masked* intensity distribution: *G*(*r*)s calculated from two of the *F*(*Q*)s in (**b**). (**d**) Effect of empirical correction when the source is the measured *masked* intensity distribution: *G*(*r*)s calculated from the other two of the *F*(*Q*)s in (**b**).

**Figure 4 nanomaterials-13-03136-f004:**
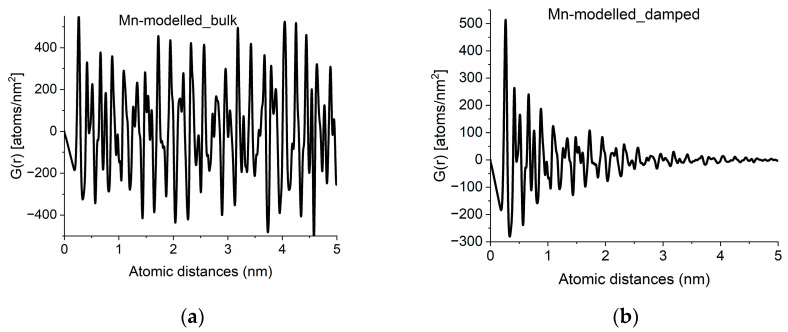
Approximation of grain-size effects by damping of *G*(*r*). (**a**) *G*(*r*) simulated for an infinite sized crystal (alpha-Mn in the example). (**b**) Damped version, to approximate grain size observed in experimental curves like Figure 2a.

**Figure 5 nanomaterials-13-03136-f005:**
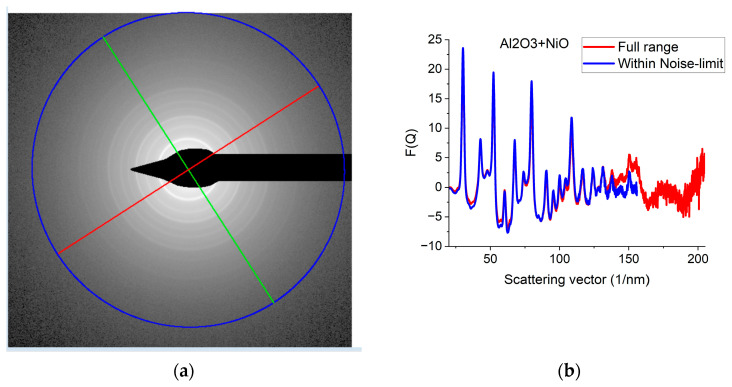
Effect of Q_max_ illustrated on Al_2_O_3_+NiO thin film sample. (**a**) 2D pattern measured from nanocrystalline Al_2_O_3_+NiO thin film, illustrating that signal buried in noise can limit useful *Q*-range. The blue circle and the two lines aid centering and are visualized optionally; (**b**) Comparison of *F*(*Q*)s calculated for the full *Q*-range of the pattern and *F*(*Q*) from the same pattern calculated from the useful *Q*-range (not dominated by noise). (**c**) Comparison of *F*(*Q*)s measured with different camera lengths (L). (**d**) Comparison of *G*(*r*)s calculated from the *F*(*Q*)s in (**c**). The *G*(*r*) functions converge to the one recorded with the longest *Q*-range (shortest camera length, L).

**Figure 6 nanomaterials-13-03136-f006:**
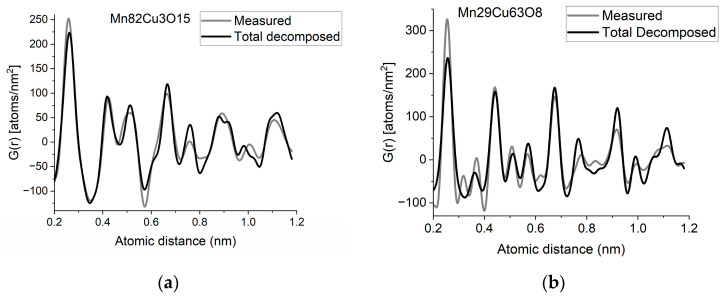
(**a**) Measured *G*(*r*) from Mn82Cu3O15 sample compared with the synthetized one, where the calculated model components (calculated for crystalline alpha-Mn, fcc-Cu and MnO) are weighted with the fractions determined from decomposition (80%Mn + 12%MnO + 8%Cu). (**b**) Measured *G*(*r*) from Mn29Cu63O8 sample compared with the synthetized one, where the calculated model components (calculated for crystalline alpha-Mn, fcc-Cu and MnO) are weighted with the fractions determined from decomposition (24%Mn + 4%MnO + 72%Cu). The true EDS compositions with oxygen are shown in the graphs. (**c**) Component PDFs obtained by decomposition are compared to the measured one for sample Mn82Cu3O15. (**d**) Component PDFs obtained by decomposition are compared to the measured one for sample Mn29Cu63O8.

**Figure 7 nanomaterials-13-03136-f007:**
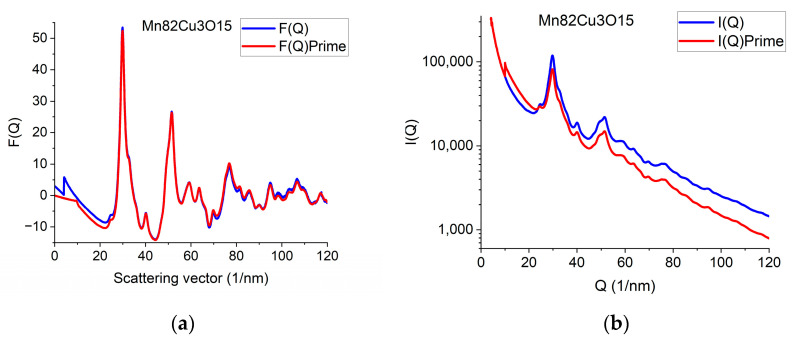
Demonstration of what changes in reciprocal space are needed to be in accord with the requirement of atomicity (and to correspond to our empirical correction). (**a**) Comparing original *F*(*Q*) and *F*’(*Q*), which satisfies atomicity and the applied empirical correction. (**b**) Measured *I*(*Q*) compared with *I*’(*Q*) that would correspond to atomicity condition and the applied empirical correction.

**Table 1 nanomaterials-13-03136-t001:** Example for results of Fingerprinting. *G*(*r*) from Mn82Cu3O15 is compared to a library of *G*(*r*)s simulated for alpha-Mn, fcc-Cu, MnO and Al_2_O_3_. (The last one (Al_2_O_3_) has nothing to do with the sample. It is used to demonstrate the effect of strongly changing Pearson’s values.) The unintentional presence of oxygen is attributed to the fact that the surface of the Mn-Cu TEM lamella must have been oxidized. That is why *G*(*r*) for MnO is also calculated and compared to the measured one.

Reference	Pearson’s Coefficient
alpha-Mn	0.948
fcc-Cu	0.579
MnO	−0.094
Al_2_O_3_	−0.191

## Data Availability

Data are contained within the article.

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
