# Peer review of "Simple ePDF: A Pair Distribution Function Method Based on Electron Diffraction Patterns to Reveal the Local Structure of Amorphous and Nanocrystalline Materials"

_nanomaterials, 2023, doi:10.3390/nano13243136_

Round 1
Reviewer 1 Report
Comments and Suggestions for Authors
1. There is a grammar issue in the title of the manuscript, please revise it carefully.
2. The abstract of the manuscript is like a piece of advertising text, without clearly stating the main scientific conclusions of the article. Please rewrite the abstract section, paying attention to writing specific technical strategies and innovations.
3. The text in all the pictures is too small for readers to read clearly. Please enlarge the text in all the Figures.
4. Line141-142, The angle bracket, notes averaging for the atomic specifications presentation It is not common to use<>to represent a function. It can be written as AVE(), or a similar representation of common functions.
5. Non original formulas in the text need to provide references.
6. After 2.9, the manuscript goes to 4 Conclusion, where is PART 3 ?
7. What is the result of a changing grain size between a. and b. in Figure 4? Is there a significant regularity between G (r) and grain size?
8. Part 2.7 is very interesting. If the PDF curves of each phase after decomposition are displayed, it will have better expressiveness.
9. Although professionals can analyze the scale of short range order and medium range order from the graphs, the manuscript should clearly provide the final order scales.
10. The article should introduce the detailed information of the program, including its components, code, error analysis, input and output results, etc. Otherwise, this article actually introduces a method, not a program.
11. The conclusion should provide a more detailed introduction to the main content and results of this article, pointing out its innovation and research significance.
Comments on the Quality of English Language1. There is a grammar issue in the title of the manuscript, please revise it carefully.
2. The abstract of the manuscript is like a piece of advertising text, without clearly stating the main scientific conclusions of the article. Please rewrite the abstract section, paying attention to writing specific technical strategies and innovations.
3. The text in all the pictures is too small for readers to read clearly. Please enlarge the text in all the Figures.
4. Line141-142, The angle bracket, notes averaging for the atomic specifications presentation It is not common to use<>to represent a function. It can be written as AVE(), or a similar representation of common functions.
5. Non original formulas in the text need to provide references.
6. After 2.9, the manuscript goes to 4 Conclusion, where is PART 3 ?
7. What is the result of a changing grain size between a. and b. in Figure 4? Is there a significant regularity between G (r) and grain size?
8. Part 2.7 is very interesting. If the PDF curves of each phase after decomposition are displayed, it will have better expressiveness.
9. Although professionals can analyze the scale of short range order and medium range order from the graphs, the manuscript should clearly provide the final order scales.
10. The article should introduce the detailed information of the program, including its components, code, error analysis, input and output results, etc. Otherwise, this article actually introduces a method, not a program.
11. The conclusion should provide a more detailed introduction to the main content and results of this article, pointing out its innovation and research significance.
Reviewer 2 Report
Comments and Suggestions for Authors
Dear authors, dear editor,
the present manuscript describes functionalities, possibilities and improvements of a new
(version of a) computer program for the evaluation of short and medium range order in
amorphous materials from electron diffraction patterns.
Overall I enjoyed reading the manuscript and I would recommend the publication
of the manuscript after some revisions/improvements
- On page 4 line 124 it is stated that the summation of the data in its 2D form
is favorable compared to the summation of the data in its 1D form. Why?
- Are there any ideas on the reason for the increasing F(Q) for high Q-values?
Possibly their origins could be discussed. Something that crossed my mind in this
respect is, that one tries to go out to high scattering angles in the experiment,
as this is required to get as as possible spatial resolution in real space (see 2.6).
In figure 1 (a) one can see the edge of the differential pump aperture.
If one measures close to its border, there an increased intensity can be found
due to the caustics, which bend back electrons scattered to higher angles into lower ones.
I am not sure, how far one measures out in practise.
- I read several time over the abstract on page 7 lines 231-237, but still did not get
how \rho_0 and r_0 are determined. One should overwork this part in order to make it
clearer how these two quantities are determined.
Typos and language issues
- A typo on page 4 line 142: eDPF -> ePDF
- on page 7 authors user the word "linear" as a noun. I am not native speaker, but I think "linear"
cannot be used as a noun and should read "linear function"
see report to authors
Round 2
Reviewer 1 Report
Comments and Suggestions for Authors
The title of the manuscript can be changed to: Simple ePDF: A PDF method based on electron diffraction patterns to reveal the local structure of amorphous and nanocrystalline materials
Comments on the Quality of English LanguageThe title of the manuscript can be changed to: Simple ePDF: A PDF method based on electron diffraction patterns to reveal the local structure of amorphous and nanocrystalline materials
Author Response
The authors thank the reviewer for careful examination of the revised version and for helping us with suggesting modification of the title. We made all changes to comply with the remarks (itemized below).
1/ The title changed as suggested by the reviewer.
2/ Chapter 2.7 was extended to give a more detailed presentation of the results.
3/ English language improved.
4/ A few small corrections done (like correction of misspelling in the legend of Figure 5.b).